# A Comparison with Adverse Events Following Immunization Associated with Sabin-Strains and Salk-Strains Inactivated Polio Vaccines in Zhejiang Province, China

**DOI:** 10.3390/vaccines10020319

**Published:** 2022-02-17

**Authors:** Huakun Lv, Xuejiao Pan, Hui Liang, Yaping Chen, Ying Wang, Fuxing Chen, Linzhi Shen, Yu Hu

**Affiliations:** Institute of Immunization and Prevention, Zhejiang Center for Disease Control and Prevention, Hangzhou 310051, China; zjmyslhk@sina.com (H.L.); xjpan@cdc.zj.cn (X.P.); hliang@cdc.zj.cn (H.L.); ypchen@cdc.zj.cn (Y.C.); ywang@cdc.zj.cn (Y.W.); fxchen@cdc.zj.cn (F.C.); lzhshen@cdc.zj.cn (L.S.)

**Keywords:** adverse events following immunization, inactivated polio vaccine, Sabin-strains, Salk-strains

## Abstract

Objectives: One dose of Sabin-strains inactivated polio vaccine (IPV) was introduced into the Chinese immunization program on 1 May 2016. This study aimed to evaluate the safety of Sabin-strains IPV and provide a comparison with conventional Salk-strains IPV. Methods: Adverse events following immunization (AEFI) records associated with Sabin-strains IPV and Salk-strains IPV were extracted from the national AEFI surveillance system (NAEFISS) from 1 May 2016 to 31 December 2020. The vaccination information on Sabin-strains IPV and Salk-strains IPV during the same period was obtained from the Zhejiang provincial immunization information system. Reporting rates of AEFI were calculated by age, city, severity of AEFI, categories of AEFI, and reaction categories and were compared between Sabin-strains IPV and Salk-strains IPV. Results: In total, 3,861,758 doses of Sabin-strains IPV and 1,018,604 doses of Salk-strains IPV were administered during the study period. The overall AEFI reporting rate for Sabin-strains IPV (3.96/10,000 doses) was significantly lower than that for Salk-strains IPV (5.03/10,000 doses) due to the reporting rate of the minor vaccine product-related reaction following Sabin-strains IPV was significantly lower than that for Salk-strains IPV (2.76/10,000 doses vs. 3.83/10,000 doses). The most frequently reported symptoms/signs were fever, induration/swelling, and rash/urticaria. The most frequently reported serious AEFI with a causal relationship was febrile convulsion, with the reporting rates of 0.10/10,000 doses for Sabin-strains IPV and 0.08/10,000 doses for Salk-strains IPV. No significant difference was found in the reporting rates of the other serious AEFI between the two types of IPV. Conclusion: Most of the AEFI following Sabin-strains IPV and Salk-strains IPV were mild and common adverse reactions. The reporting rate of serious AEFI was not significantly different between Sabin-strains IPV and Salk-strains IPV. Sabin-strains IPV had a favorable safety profile and could be widely used.

## 1. Introduction

The strategy of switching from the trivalent oral live attenuated polio vaccine (tOPV) to the bivalent OPV (bOPV), withdrawing the type II strain, was recommended by the Polio Eradication and Endgame Strategic Plan 2013–2018. Simultaneously, the World Health Organization (WHO) also recommended that OPV-using countries should arrange at least one dose of inactivated polio vaccine (IPV) into the routine polio vaccine schedule to prevent the possible outbreaks of polio virus type II or the circulation of vaccine-derived polio virus type II [1]. Based on these strategies, China switched from using tOPV to using bOPV in the routine immunization program and introduced one dose of IPV into the polio vaccine schedule on 1 May 2016, which was synchronized with more than 150 other countries using OPV in their national immunization programs [2]. The new Chinese polio vaccine schedule included one IPV dose given at 2 months of age, followed by three bOPV doses at 3, 4, and 48 months of age [3].

The China drug administration approved two types of IPV [3]. One was manufactured from Salk-strains and had been available as a voluntary vaccine subject to payment since 2009. Another was manufactured from Sabin-strains and was licensed in 2015. Salk-strains IPV or Salk-strains containing combination vaccines have been widely used in the USA and other developed countries and their safety and effectiveness have been proved [4,5]. However, Salk-strains IPV could not be widely used in many developing countries due to its high biosafety risks during production and the limited production capacity and its relevant high cost [6]. WHO encouraged the research and development of new types of IPV using less virulent virus strains, such as Sabin-strains IPV [7]. Up to date, only China and Japan license and use Sabin-strains IPV.

Immunogenicity and safety of Sabin-strains IPV and the Sabin-strains IPV-OPV sequential schedule have been evaluated by clinical trials [8,9,10]. However, the clinical trials were not powered sufficiently to detect the infrequent or rare adverse events following immunization (AEFI) due to the limited sample size. Continuous assessment of the safety analysis of the post-licensure vaccines can provide a tool to evaluate the benefit–risk profiles of a specific vaccine which cannot be evaluated in pre-licensure clinical trials. Additionally, post-licensure AEFI surveillance enables the detection of signals that will generate hypotheses, as well as the identification and rectification of gaps in this program to strengthen the routine vaccination [11,12]. Before the switch of polio vaccine schedule, only a few pilot studies (replaced the first dose of tOPV with Sabin-strains IPV) reported the results on the post-licensure AEFI surveillance of Sabin-strains IPV [13,14,15].

We also know that limited data were available regarding the safety of Sabin-strains IPV after its licensure, and consequently, during its large-scale use. Continuous surveillance on the safety of Sabin-strains IPV based on a large population helped to capture the rare AEFI or the unexpected AEFI, which could evaluate the safety profile of Sabin-strains IPV in a comprehensive manner.

Our study aimed to assess the safety of Sabin-strains IPV. Specifically, all AEFI associated with Sabin-strains IPV in Zhejiang province were summarized from the national passive surveillance data collected in the first 4.5 years (from 1 May 2016 to 31 December 2020) after the introduction of Sabin-strains IPV into the routine immunization program, with a comparison with the Salk-strains IPV during the same period. This study provides a reference for the widespread use of Salk-strains IPV in the developing countries at the endgame of polio eradication.

## 2. Materials and Methods 

### 2.1. Study Area

Zhejiang is a developed province with a large population of 70 million people in eastern areas of China. Of the total population, 7.23% children were under 7 years of age. Zhejiang province launched the EPI in 1978 with four vaccines and it continued to increase the number of vaccines up to 11 to date, with the administration of 20 million doses of vaccines each year.

### 2.2. Product Information

The poliovirus strains used by Salk-IPV are wild-type polioviruses including Mahoney (type 1), MEF-1 (type 2), and Saukett (type 3). These virus strains are inoculated in Vero cells and inactivated with formalin. Every dose of Salk-IPV contains the D-antigens of 40DU, 8DU, and 32DU for type 1, 2, and 3, respectively. Other ingredients include 2-phenoxyethanol, ethanol, methanol, sodium hydroxide, hydrochloric acid, neomycin, streptomycin, and polymyxin B. The final product is packaged with a prefilled single syringe and is valid for 36 months if it is stored under the condition of 2–8 °C.

The poliovirus strains used by Sabin-IPV are Sabin virus type 1 and type 2 and Pfizer (type 3). These virus strains are inoculated in Vero cells and inactivated with formalin. Every dose of Sabin-IPV contains the D-antigens of 30DU, 32DU, and 45DU for type 1, 2, and 3, respectively. Other ingredients include 2-phenoxyethanol, sodium hydroxide, hydrochloric acid, neomycin, and streptomycin. The final product is packaged in a Cilin bottle as one dose per vial and is valid for 24 months if it is stored under the condition of 2–8 °C.

### 2.3. Polio Vaccines Schedule in Zhejiang Province

Since 1 May 2016, one dose of Sabin-strains IPV has been included in the Zhejiang province routine immunization program to be given at 2 months of age. On 1 November 2019, two doses of Sabin-strains IPV were included to be given at 2 months and 3 months of age, respectively. The Sabin-strains IPV was funded by the government and provided to all eligible children. However, every eligible child also had a supplementary option to choose the self-paid Salk-strains IPV to replace the two doses of Sabin-strains IPV and even the following two doses of bOPV. The alternative schedule of the Salk-strains IPV included four doses at 2, 3, 4, and 18 months of age.

### 2.4. National Adverse Events following Immunization Surveillance System

The national adverse events following immunization surveillance system (NAEFISS) was an official immunization safety surveillance system that was established by the Chinese Center for Disease Control and Prevention based on WHO guidance [16]. NAEFISS aims to detect new, unusual, or rare AEFI, to evaluate the safety of newly licensed vaccines, to identify potential risk factors for AEFI, to monitor increases in known AEFI, to determine the possible reporting clusters, and to provide a reliable safety monitoring system. Zhejiang province joined the NAEFISS and started systematic surveillance in 2009. NAEFISS was upgraded in 2012 for adding variables of the case reporting form and rules of data logic verification.

### 2.5. AEFI Reporting and Investigation Procedures

The national AEFI guidance was released in 2010 [16]. An AEFI case is defined as a reaction or an event occurring after vaccine administration that is suspected to be related to the vaccination. According to the national AEFI guidance, healthcare facilities, vaccination clinics, CDC at each administrative level, adverse drug reaction monitoring agencies, and vaccine manufacturers are required to report AEFI cases. Additionally, public members or guardians or parents can also notify any of the authorized reporters mentioned above to report an AEFI case.

After identifying an AEFI case, all the above-authorized reporters should report it to the vaccination clinic or the county-level CDC within the jurisdiction. The vaccination clinic or the county-level CDC then completes an “AEFI Case Reporting Card” and submits the data to NAEFISS. The investigation is required for all AEFI, except for the non-serious vaccine product-related reaction with a clear diagnosis (e.g., fever, erythema, swelling, and induration at the injection site). If necessary, an AEFI case is investigated by the county-level CDC within the jurisdiction. In case of deaths, serious AEFI, AEFI clusters, and AEFI of significant public concern that are suspected to be related to vaccination, prefectural or provincial within the jurisdiction expert committees (comprising pediatricians, physicians, epidemiologists, pharmacists, vaccine researchers, etc.) are responsible for further investigation and causality assessment.

The reporting form or the investigation collects the information on the vaccinated individual, storage and transportation of vaccines, vaccine administration, and the AEFI itself. Signs and symptoms of AEFI are coded using the International Classification of Diseases (version 10.0, ICD-10), a clinically validated, internationally standardized terminology. A single AEFI report may be assigned more than one term and be referred to more than one suspected vaccine. 

### 2.6. AEFI Category

An AEFI is any adverse medical occurrence that follows immunization, but which does not necessarily have a causal relationship with vaccination. According to the cause-specific classification of AEFI from WHO [17], AEFI cases are divided into five types: (1) vaccine product-related reaction (including non-serious reaction and serious reaction); (2) vaccination error (including errors in vaccine handling, errors in vaccine prescribing or non-adherence to recommendations for use, and errors in administration); (3) vaccine quality defect-related reaction; (4) coincidental event; and (5) anxiety reaction.

According to the national AEFI guidance [16], a serious AEFI is defined as an event that results in death, is life-threatening, requires hospitalization or prolongs the existing hospitalization, results in persistent or significant disability, causes a congenital anomaly or birth defect, or requires intervention to prevent permanent impairment or damage. For example, serious AEFI include but are not limited to allergic shock, allergic laryngeal edema, allergic purpura, thrombocytopenic purpura, Arthus reaction, febrile convulsion, epilepsy, brachial neuritis, polyneuritis, Guillain–Barre syndrome, encephalopathy, encephalitis and meningitis, syncope, etc.

### 2.7. Data Source

The vaccination records of Sabin-strains IPV and Salk-strains IPV (standalone only, combination vaccines were not included) administered from 1 May 2016 to 31 December 2020 in Zhejiang province were extracted from the Zhejiang provincial immunization information system (ZJIIS), which was established and maintained by Zhejiang Provincial Center for Disease Control and Prevention (ZJCDC) to collect and manage the information on vaccine procurement and distribution, vaccination clinic information, and cold chain temperature information. The function of ZJIIS can be found elsewhere [18]. Each recipient’s vaccination information is submitted to ZJIIS by vaccination staff.

AEFI records following Sabin-strains IPV and Salk-strains IPV administered from 1 May 2016 to 31 December 2020 in Zhejiang province were obtained from NAEFISS on 1 April 2021 to account for data lags due to the correction and cleaning.

### 2.8. Outcome and Data Analysis

A database of the AEFI records associated with Sabin-strains IPV and Salk-strains IPV administration during the study period was organized as an Excel file (Microsoft Office Excel 2020). The characteristics of AEFI records were summarized by gender of case, causal category, type of reporter, severity, city, patient age, interval of AEFI onset (from vaccination date (day 0) to onset of first symptoms), and symptoms/signs/diagnoses. Each AEFI record might list several symptoms, signs, and/or diagnoses, but only the main symptom or the most serious diagnosis was included.

The reporting rates of the general AEFI and the serious AEFI were calculated for Sabin-strains IPV and Salk-strains IPV, respectively. The reporting rate was calculated by dividing the number of AEFI reports by the relevant vaccine doses administered. We graphically depicted monthly reporting rates during the study period to display its trends. Some of the serious AEFI following Sabin-IPV and Salk-IPV were summarized by the confirmed causal relationship.

The Pearson chi-square test or Fisher’s exact test was used to examine differences in reporting rate between Sabin-strains IPV and Salk-strains IPV. If at least one theoretical frequency of the cross-table was <5, Fisher’s exact test was used. Otherwise, the Pearson chi-square method was used. Hypothesis testing was two-sided, and *p*-values < 0.05 were considered to be statistically significant. All statistical analyses were performed using Stata 14.0 (Stata Corp. 2015, Stata Statistical Software, College Station, TX, USA).

## 3. Results

In total, 3,861,758 doses of Sabin-strains IPV and 1,018,604 doses of Salk-strains IPV were administered from 1 May 2016 to 31 December 2020 in Zhejiang province, respectively. Children using the sequential vaccination schedule for Sabin-strains IPV and Salk-strains IPV were not found in this study. All of the children who chose the Salk-strains IPV were only administered with Salk-strains IPV for the entire schedule. Subsequently, a total of 1528 AEFI cases associated with Sabin-strains IPV (reporting rate: 3.96/10,000 doses) and 513 AEFI cases associated with Salk-strains IPV (reporting rate: 5.04/10,000 doses) were reported. The general AEFI reporting rate for Sabin-strains IPV was significantly lower than that for Salk-strains IPV (*p* < 0.05). The monthly reporting rates for Sabin-strains IPV varied at a relatively stable rate, from 0.95 to 8.70 per 10,000 doses. However, the monthly reporting rate of Salk-IPV had three peaks during the study period in June 2017, June 2018, and November 2020, with a variation from 0.00 to 46.35 per 10,000 doses. The general AEFI reporting rate was slightly higher for Salk-strains IPV than that for Sabin-strains IPV (*p* < 0.05). In total, 116 serious AEFI cases following Sabin-strains IPV and 33 serious AEFI cases following Salk-strains IPV were reported, and no significant difference was observed in the reporting rate of AEFI for the two types of IPV (0.30/10,000 doses for Sabin-strains IPV and 0.32/10,000 doses for Salk-strains IPV). The reporting rate of serious AEFI was not significantly different between the two types of IPV (Figure 1).

Table 1 presents the characteristics of the AEFI records following the two types of IPV. For both Sabin-strains IPV and Salk-strains IPV, the majority of AEFI was minor vaccine product-related reactions and the reporting rate for Salk-strains IPV was significantly higher than that for Sabin-strains IPV (2.76/10,000 doses vs. 3.83/10,000 doses, *p* < 0.01). No significant difference was observed in the reporting rates of severe vaccine product-related reactions (0.53/10,000 doses vs. 0.50/10,000 doses, *p* > 0.05), vaccination errors (0.01/10,000 doses vs. 0.01/10,000 doses, *p* > 0.05), and coincidental events (0.65/10,000 doses vs. 0.70/10,000 doses, *p* > 0.05). No AEFI reports were classified as anxiety-related reactions.

A trend of a higher reporting rate was observed in the subsequent doses (e.g., the reporting rate of the fourth dose of Sabin-strains IPV was 7.88/10,000 doses and was 6.04/10,000 doses for the fourth dose of Salk-strains IPV, *p* < 0.05). The majority reports came from healthcare providers (97.91% for Sabin-strains IPV and 96.88% for Salk-strains IPV). The reporting rate onset within 48 h after Sabin-strains IPV vaccination was 1.94/10,000 doses, while it was significantly higher for Salk-strains IPV (2.97/10,000 doses). The reporting rate of AEFI following Sabin-strains IPV administered standalone was 2.48/10,000 doses, while it was significantly lower for Salk-strains IPV (0.46/10,000 doses).

For both two types of IPV, the top three most frequently reported symptoms and/or signs were fever (1.56/10,000 doses for Sabin-strains IPV and 2.03/10,000 doses for Salk-strains IPV), induration/swelling (1.01/10,000 doses for Sabin-strains IPV and 1.52/10,000 doses for Salk-strains IPV), and rash/urticaria (0.95/10,000 doses for Sabin-strains IPV and 1.00/10,000 doses for Salk-strains IPV). Fever and induration/swelling were reported at a significantly higher rate following Salk-strains IPV administration compared to Sabin-strains IPV (*p* < 0.05). No significant difference was observed in the reporting rate for other symptoms and/or signs between the two types of IPV (Table 2).

After the causality assessment, the most frequently reported serious clinical diagnosis was febrile convulsion (0.10/10,000 doses for Sabin-strains IPV and 0.08/10,000 doses for Salk-strains IPV). Thirty AEFI cases were classified as related to vaccination with Sabin-strains IPV, including six allergic purpura cases (reporting rate 0.02/10,000 doses), eight thrombocytopenia cases (reporting rate 0.02/10,000 doses), four angioedema cases (reporting rate 0.01/10,000 doses), and eleven febrile convulsion cases (reporting rate 0.10/10,000 doses). Nine AEFI cases were classified as related to vaccination with Salk-strains IPV, including two allergic purpura cases (reporting rate 0.02/10,000 doses), two thrombocytopenia cases (reporting rate 0.02/10,000 doses), two angioedema cases (reporting rate 0.02/10,000 doses), and three febrile convulsion cases (reporting rate 0.08/10,000 doses). The other serious AEFI associated with two types of IPV were identified as coincidental events. No statistically significant difference was found in the reporting rates of the other serious AEFI with the causal relationship between the two types of IPV (Table 3).

## 4. Discussion

This study presented an evaluation of AEFI of the domestic Sabin-strains IPV through the national passive surveillance system on AEFI in Zhejiang province, with a time span of 4.5 years after the switch of polio vaccination strategy. Furthermore, this study also compared the reporting rates of AEFI following the new licensed Sabin-strains IPV to those of the conventional Salk-strains IPV.

The overall AEFI reporting rates for Sabin-strains IPV and Salk-strains IPV were 3.96 and 5.04 per 10,000 doses, respectively. Compared with the previous AEFI surveillance data of the domestic Sabin-strains IPV, the overall AEFI reporting rate of Sabin-strains IPV observed in this study was much lower than those from two pilot studies. For example, Fu [13] reported the AEFI rate of Sabin-strains IPV was 16.89/10,000 doses through the passive surveillance and Shi [14] found the AEFI rate of Sabin-strains IPV was 246.43/10,000 doses through the active surveillance. There were two possible explanations for this difference. One was that the active AEFI surveillance was carried out during the pilot period. The sensitivity of active monitoring was much higher than that of the routine passive surveillance. Another was that a new vaccine would attract a high social concern of its safety as reporters could have been more likely to report more AEFI cases at the very beginning of its use. Similar rises in reporting rates following the vaccine introduction had been reported as a phenomenon of the Weber effect [19]. It described an increase in the reporting rates of adverse events occurring in the time period shortly after the drug marketing, followed by a stabilization. Except for the two reports mentioned above, the overall AEFI reporting rate for Sabin-strains IPV in this study is higher than that reported by Wang [15] (2.69/10,000 doses), which was based on the passive surveillance. We assumed that it might be associated with the disparities in the study design, such as reporting sensitivity, sample size, and observation period.

Salk-strains IPV has been widely used in the developed countries usually in the form of a combination vaccine (e.g., DTaP-Hib/IPV) [20,21]. Only a few AEFI surveillance reports were associated with the vaccination of standalone Salk-strains IPV. The overall AEFI reporting rate of the standalone Salk-strains IPV ranged from 0.01 to 0.11 per 10,000 doses in some developed countries [22,23,24]. However, the AEFI reporting rate of Salk-strains IPV in this study was much higher than those from the previous reports. We assumed that the variation in the AEFI reporting rate following Salk-strains IPV in different countries might be explained by the variable reporting requirements, case definitions, and surveillance sensitivity. The AEFI reporting rate for Salk-strains IPV observed in our study was also higher than that reported by Li [25] (1.6/10,000 doses) in the analysis of the nationwide AEFI surveillance through the NAEFISS. However, this report used the number of Salk-strains IPV distribution as the denominator for calculating the reporting rate of AEFI. It would underestimate the AEFI reporting rate as the number of distributed doses was always greater than the actual number of administered doses. Furthermore, the surveillance sensitivity in Zhejiang province was higher than that of the other regions of China and the national average level [25].

In our study, we found the AEFI reporting rate of Salk-strains IPV was slightly higher than that of Sabin-strains IPV. Further analysis indicated that the difference between two types of IPV was mainly due to the significantly higher reporting rate of the minor vaccine product-related reactions for Salk-strains IPV. The differences between two types of IPV in the reporting rate of severe vaccine product-related reactions, vaccination errors, and coincidental events were not significant.

Consistent with the previous reports, we found the most common AEFI cases for two types of IPV were classified as the minor vaccine product-related reactions, which were relatively minor, mild, and self-limited [13,14,15]. Fever was the most frequent symptom and/or sign observed in AEFI cases associated with both Sabin-strains IPV and Salk-strains IPV. It was in line with the results from surveillance reports and clinical trials [13,14,15,25]. One phase III clinical trial of Sabin-strains IPV found that the incidence of fever was more frequent in the Sabin-strains IPV group than that in the Salk-strains IPV group [18]. On the contrary, our result was opposite to these two pieces of evidence mentioned above, with a higher reporting rate observed in Salk-strains IPV. The specific reason for this phenomenon was yet unknown, but it might be related to different production processes. Induration/swelling at injection site was the second most frequently reported symptom associated with both two types of IPV, which was similar to the results from several previous reports as the most common local symptom [18,26]. Both two types of IPV have the aluminum adjuvant, which had been reported that it could induce the redness and induration [27,28]. We also found the reporting rate of induration/swelling was significantly higher following Salk-strains IPV. It might be explained by the different dosages of adjuvant or the different syringes and needles used in two types of IPV vaccination. Pre-filled syringes produced by Becton, Dickinson, and Company (BD) were used to inject Salk-IPV, while common syringes and needles were used to inject Sabin-IPV. However, it needs to be further investigated. Rash/urticaria, which was a common clinical manifestation of the hypersensitivity reaction, was the third most frequently reported AEFI following two types of IPV. It was consistent with the results from other AEFI surveillance studies [29,30]. The main reason would be due to the allergy of the recipient to the components of the vaccine, such as the proteins.

After the causality assessment, the proportion of serious AEFI that had a causal relationship was very small in both two types of IPV, as well as the relevant reporting rates. Generally, the reporting rate of the serious AEFI was very low associated with Sabin-strains IPV and Salk-strains IPV. Sabin-strains IPV had a safety profile consistent with that of Salk-strains IPV [7]. Febrile convulsion was the most frequently reported serious AEFI that was found to be related to both Sabin-strains IPV and Salk-strains IPV vaccination. A febrile convulsion is defined as a seizure occurring in a child 6 months to 5 years old that is accompanied by a fever (≥38 °C) without central nervous system infection [31]. Febrile convulsion can be induced under the stress of a fever and vaccination has been found to be the second most common cause. However, controversy on the association between vaccination and febrile convulsion still exists [32]. Another finding for the causality assessment of the serious AEFI reports was that the consistency for Sabin-IPV was not as good as that for Salk-IPV. The possible explanation was that the reporting sensitivity of the serious AEFI following Sabin-IPV was higher due to the doubts on safety of the new vaccines from the public. However, most of the reported serious AEFI following Sabin-IPV were considered as coincidental events after rigorous evaluation on causal relationships.

There were still several limitations regarding this study. First, NAEFISS, as a passive surveillance system, has some inherent disadvantages, such as reporting bias (under-reporting or over-reporting), inconsistency and completeness of reports, and a lack of control groups. Second, the description of mild signs and symptoms without medical checks mainly relied on the recollections of parents or caregivers. This information was not verified by health professionals. Third, the standard definition of AEFI diagnosis, which was created by the Brighton Collaboration [33] and widely used internationally, was not applied in our study as the national AEFI guidance did not use the definitions of AEFI from the Brighton Collaboration. There would be obstacles for comparing our findings to that from other studies.

## 5. Conclusions

This study indicated that most of the AEFI following Sabin-strains IPV and Salk-strains IPV were mild and common adverse reactions. The reporting rate of serious AEFI that had a causal relationship with IPV was very low and no difference was observed between Sabin-strains IPV and Salk-strains IPV. The reporting rate of the minor vaccine product-related reactions following Salk-strains IPV was slightly higher but within a reasonable range. Therefore, we assumed that the Sabin-strains IPV had a favorable safety profile that was similar to the Salk-strains IPV. A polio-free world needs the improved safety levels in vaccine production facilities. As such, the Sabin-strains IPV could be widely used, which could contribute to achieving the polio eradication target in developing countries and maintaining a polio-free world.

## Figures and Tables

**Figure 1 vaccines-10-00319-f001:**
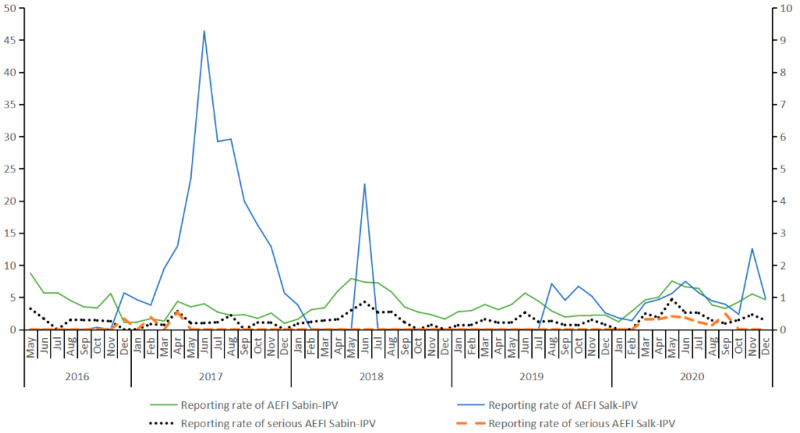
The reporting trends of AEFI following inactivated polio vaccines from 2016 to 2020 (reporting rate: /10,000 doses).

**Table 1 vaccines-10-00319-t001:** The reporting rate of AEFI following Sabin-IPV and Salk-IPV from 1 May 2016 to 31 December 2020, Zhejiang province.

Variable	Level	Sabin-IPV				Salk-IPV				*p* ^※^
		Administered Doses	No. of AEFI Case	Reporting Rate *	*p ^#^*	Administered Doses	No. of AEFI Case	Reporting Rate *	*p ^#^*	
Gender	Male	1,930,234	761	3.94	>0.05	504,485	253	5.02	>0.05	<0.05
	Female	1,931,524	767	3.97		514,119	260	5.06		<0.05
Dose number	1	2,161,210	527	2.44	<0.01	269,751	113	4.19	<0.01	<0.01
	2	802,665	313	3.90		216,944	92	4.24		>0.05
	3	85,373	48	5.62		121,467	60	4.94		<0.05
	4	812,510	640	7.88		410,442	248	6.04		<0.05
Category ^★^	Vaccine product-related reaction (minor)	3,861,758	1065	2.76	<0.01	1,018,604	390	3.83	<0.01	<0.01
	Vaccine product-related reaction (severe)	3,861,758	206	0.53		1,018,604	51	0.50		>0.05
	Vaccination error	3,861,758	5	0.01		1,018,604	1	0.01		>0.05
	Coincidental event	3,861,758	252	0.65		1,018,604	71	0.70		>0.05
Severity	Serious	3,861,758	116	0.30	<0.01	1,018,604	33	0.32	<0.01	>0.05
	Non-serious	3,861,758	1412	3.66		1,018,604	480	4.71		<0.05
Type of reporter	Healthcare provider	3,861,758	1496	3.87	<0.01	1,018,604	497	4.88	<0.01	<0.01
	Caregivers	3,861,758	30	0.08		1,018,604	16	0.16		<0.05
	Manufacturer	3,861,758	2	0.01		1,018,604	0	0.00		>0.05
Onset from vaccination	0–1 d	3,861,758	749	1.94	<0.01	1,018,604	303	2.97	<0.01	<0.05
	2–3 d	3,861,758	461	1.19		1,018,604	126	1.24		>0.05
	4–7 d	3,861,758	193	0.50		1,018,604	55	0.54		>0.05
	8–14 d	3,861,758	66	0.17		1,018,604	18	0.18		>0.05
	≥15 d	3,861,758	59	0.15		1,018,604	11	0.11		>0.05
Year of onset	2016	278,711	127	4.56	>0.05	321,970	18	0.56	<0.01	<0.01
	2017	605,775	142	2.34		159,410	223	13.99		<0.01
	2018	709,006	310	4.37		13,608	3	2.20		<0.01
	2019	942,564	304	3.23		37,535	17	4.53		<0.05
	2020	1,325,702	645	4.87		486,081	252	5.18		>0.05
Administered alone		962,239	239	2.48		2,538,033	117	0.46		<0.01

* /10,000 doses. ^#^ Comparison of the reporting rates for the variable level in the group of Sabin-IPV or in the group of Salk-IPV. ^※^ Comparison between the groups of Sabin-IPV and Salk-IPV. ^★^ Vaccine product-related reaction (minor) was a reaction of physiological dysfunction, included fever, local swelling, tiredness, loss of appetite, fatigue, etc. Vaccine product-related reaction (severe) was a reaction of impairment of body tissues, organs, and functions.

**Table 2 vaccines-10-00319-t002:** Symptoms/signs/diagnosis of AEFI following Sabin-IPV and Salk-IPV from 1 May 2016 to 31 December 2020, Zhejiang province.

AEFI Symptoms and/or Signs	Sabin-IPV			Salk-IPV			*p* ^#^
	No. of Cases	%	Reporting Rate *	No. of Cases	%	Reporting Rate *	
Fever	602	39.40	1.56	207	40.35	2.03	<0.05
Persistent crying	52	3.40	0.13	19	3.70	0.19	>0.05
Induration/swelling	389	25.46	1.01	155	30.21	1.52	<0.05
Vomiting	25	1.64	0.06	6	1.17	0.06	>0.05
Diarrhea	5	0.33	0.01	0	0.00	0.00	>0.05
Rash/urticaria	367	24.02	0.95	102	19.88	1.00	>0.05
Allergic purpura	22	1.44	0.06	2	0.39	0.02	>0.05
Thrombocytopenia	11	0.72	0.03	2	0.39	0.02	>0.05
Anaphylactic shock	1	0.07	0.00	0	0.00	0.00	>0.05
Angioedema	4	0.26	0.01	3	0.58	0.03	>0.05
Febrile convulsion	42	2.75	0.11	15	2.92	0.15	>0.05
Guillain–Barre Syndrome	2	0.13	0.01	1	0.19	0.01	>0.05
Acute disseminated encephalomyelitis	1	0.07	0.00	0	0.00	0.00	>0.05
Epilepsy	1	0.07	0.00	0	0.00	0.00	>0.05
Encephalitis and meningitis	3	0.20	0.01	1	0.19	0.01	>0.05
Encephalopathy	1	0.07	0.00	0	0.00	0.00	>0.05

* /10,000 doses. ^#^ Comparison between reporting rates of AEFI following Sabin-IPV and Salk-IPV.

**Table 3 vaccines-10-00319-t003:** Some of the serious AEFI following Sabin-IPV and Salk-IPV from 1 May 2016 to 31 December 2020, Zhejiang province.

Clinical Diagnosis	Sabin-IPV			Salk-IPV			*p ^#^*
	No. of Cases	Consistent	Reporting Rate *	No. of Cases	Consistent	Reporting Rate *	
Allergic purpura	22	6	0.02	2	2	0.02	>0.05
Thrombocytopenia	11	8	0.02	2	2	0.02	>0.05
Anaphylactic shock	1	0	0.00	0	0	0.00	>0.05
Angioedema	4	4	0.01	3	2	0.02	>0.05
Febrile convulsion	42	11	0.10	15	3	0.08	>0.05
Guillain–Barre Syndrome	2	0	0.00	1	0	0.00	>0.05
Acute disseminated encephalomyelitis	1	0	0.00	0	0	0.00	>0.05
Epilepsy	1	0	0.00	0	0	0.00	>0.05
Encephalitis and meningitis	3	0	0.00	1	0	0.00	>0.05
Encephalopathy	1	0	0.00	0	0	0.00	>0.05

* /10,000 doses. ^#^ Comparison between reporting rates of AEFI following Sabin-IPV and Salk-IPV.

## Data Availability

The data could be found on the national AEFI surveillance system which was not open to the public.

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
