# Peer review of "A Comparison with Adverse Events Following Immunization Associated with Sabin-Strains and Salk-Strains Inactivated Polio Vaccines in Zhejiang Province, China"

_vaccines, 2022, doi:10.3390/vaccines10020319_

Round 1
Reviewer 1 Report
This is an interesting report: comparison of AEFIs associated with Salk- and Sabin-type IPV vaccines given to children in a large province of China (3 million doses vs 1 million administered doses ). Methods are correct, some technical info is missing. Authors are invited to improve their MS as follows:
Abstract: Insert the size of the investigated populations receiving either Salk or Sabin standalone vaccines. Also specify whether children did receive 1 or more doses of each vaccine (this issue is unclear in the abstract, text and Table 1). Likely, most children received only 1 dose, some of them received 2,3, or 4 doses. Please note that the way used by MDPI journals for formatting Tables is probably not optimal for readers: 1) items and numbers are centered in columns (if numbers were aligned to the right side, differences would be more evident); 2) numbers are formatted without using thousands separators. This needs to be corrected when we are dealing with 5-6 digits. This note is sent also to the Editor.
Line 47: "private" vaccine; probably should be referred to as "voluntary vaccine subject to payment"
Line 67: we also know
Materials and Methods:
A section should added: Salk and Sabin IPV vaccines - Product information
Important info to be inserted:
Virus strains present in vaccine (e.g., Salk: Mahoney type 1, MEF-1 (Middle East Forces) type 2 and. Saukett type 3 - Sabin: ...). Of course, actual strains used by China producers.
Inactivating agent (name, concentration, time, temperature of treatment)
Amount of virus for each strain (AgD)
Cell lines in which viruses are produced
Presence of preservative and type
Residual antibiotics and name(s)
Host component proteins (and serum proteins, if serum is used; or proteins other than those in animal serum): max tolerated amounts
Adjuvant: type and dose
Production process: concentration, filtration, sterility checks
Packaging sheet: borosilicate vials single or multiple doses; diluent (if present); prefilled single syringes; other
Storage conditions and duration (months/years from manufacturing to expiration date)
Fig. 1 and results line 181 etc: please elaborate on strong periodic variation of reported AEFIs
Table 1: thousands separators are needed + numbers aligned to the right.
Line 209 etc: are data relating ONLY to standalone IPV or data of combined vaccines are also include? This would change interpretations and needs to be made clear.
Line 304: different types/brands of syringe/needles used. This can be one cause of AEFIs. Are there specific qualitative requirements for syringes/needles? Are these respected? Please elaborate (here costs are a dominant issue).
Lines 337: a sentence may be added to the last line: polio-free world with improved safety levels in vaccine production facilities.
Literature is OK. Since polio literature is emerging continuosly, please take care to re-check references accordingly.
Author Response
Abstract: Insert the size of the investigated populations receiving either Salk or Sabin standalone vaccines. Also specify whether children did receive 1 or more doses of each vaccine (this issue is unclear in the abstract, text and Table 1). Likely, most children received only 1 dose, some of them received 2,3, or 4 doses. Please note that the way used by MDPI journals for formatting Tables is probably not optimal for readers: 1) items and numbers are centered in columns (if numbers were aligned to the right side, differences would be more evident); 2) numbers are formatted without using thousands separators. This needs to be corrected when we are dealing with 5-6 digits. This note is sent also to the Editor.
Response:The size of target population was added in the abstract. The format of tables and numbers were revised. Children using the sequential vaccination schedule for Sabin-strains IPV and Salk-strains IPV were not found in this study. All of the children who chose the Salk-strains IPV were only administered with Salk-strains IPV for the entire schedule.
Line 47: "private" vaccine; probably should be referred to as "voluntary vaccine subject to payment"
Response: we revised this sentence.
Line 67: we also know
Response: we revised this sentence.
Materials and Methods:
A section should added: Salk and Sabin IPV vaccines - Product information
Important info to be inserted:
Virus strains present in vaccine (e.g., Salk: Mahoney type 1, MEF-1 (Middle East Forces) type 2 and. Saukett type 3 - Sabin: ...). Of course, actual strains used by China producers.
Inactivating agent (name, concentration, time, temperature of treatment)
Amount of virus for each strain (AgD)
Cell lines in which viruses are produced
Presence of preservative and type
Residual antibiotics and name(s)
Host component proteins (and serum proteins, if serum is used; or proteins other than those in animal serum): max tolerated amounts
Adjuvant: type and dose
Production process: concentration, filtration, sterility checks
Packaging sheet: borosilicate vials single or multiple doses; diluent (if present); prefilled single syringes; other
Storage conditions and duration (months/years from manufacturing to expiration date)
Response: we added a production part in the methods section. All the information was from the package insert of the products, but some of the information was not found in the package inserts of both two IPVs.
Fig. 1 and results line 181 etc: please elaborate on strong periodic variation of reported AEFIs
Response: we elaborated the results.
Table 1: thousands separators are needed + numbers aligned to the right.
Response: we revised the tables.
Line 209 etc: are data relating ONLY to standalone IPV or data of combined vaccines are also include? This would change interpretations and needs to be made clear.
Response: The Salk-IPV containing combined vaccine were not included in our study.
Line 304: different types/brands of syringe/needles used. This can be one cause of AEFIs. Are there specific qualitative requirements for syringes/needles? Are these respected? Please elaborate (here costs are a dominant issue).
Response: Pre-filled syringes produced by Becton, Dickinson, and Company (BD) were used to inject Salk-IPV, while common syringes and needles were used to inject Sabin-IPV.
Lines 337: a sentence may be added to the last line: polio-free world with improved safety levels in vaccine production facilities.
Response: Thank you for your suggestion.
Literature is OK. Since polio literature is emerging continuosly, please take care to re-check references accordingly.
Response: we updated some of the references.
Reviewer 2 Report
1.This study was secondary data analysis based on the from May 1, 2016 to Dec 31, 2020 in Zhejiang province. The authors must describe briefly about the guidelines for administering IPV vaccine. Which type of IPV vaccines were given for which population in Zheijiang province? Is it randomly or based on the supply from government (eg Salk strain in 2016 and Sabin strain in 2017, etc)
2.Regarding the reporting of AEFI events, both vaccines were reported by same medical personnel or study areas are different place ?
- According to Figure-1, we can see three peaks of Salk vaccine reporting rate. Why there was the peak for reporting rate? Is there any reasons or briefly discuss about it.
- In conclusion, the authors stated that the reporting rate and occurrence of AEFI for both vaccines were not different and the authors stated that Sabin vaccine can use safely. Why the authors describe Sabin only and not for Salk vaccine? Is there any reasons or check your conclusions and revise it.
5.At Table-1 , the authors categorized vaccine product related reaction (minor and severe),. How did you classify and please add the operational definition as foot note in the table.
- At table -1 the authors described vaccination error. What is the meaning of vaccination error? Please add operational definition at the revised manuscript.
7.At table-3, the reporting of serious AEFI event of Sabin vaccine was higher and but the consistent was not good. Is it due to over interpretation or any other reasons? Please discuss it.
Author Response
1.This study was secondary data analysis based on the from May 1, 2016 to Dec 31, 2020 in Zhejiang province. The authors must describe briefly about the guidelines for administering IPV vaccine. Which type of IPV vaccines were given for which population in Zheijiang province? Is it randomly or based on the supply from government (eg Salk strain in 2016 and Sabin strain in 2017, etc)
Response: we made a statement in the methods section 2.2 to describe the policy and schedule of the Salk strain- and Sabin strain-IPV. “Since May 1, 2016, one dose of Sabin-strains IPV has been included in Zhejiang province routine immunization program to be given at 2 months of age. On Nov 1, 2019, two doses of Sabin-strains IPV were included to be given at 2 months and 3 months of age, respectively. The Sabin-strains IPV was funded by the government and provided to all eligible children. However, every eligible child also had a supplementary option to choose the self-paid Salk-strains IPV to replace the two doses of Sabin-strains IPV and even the following two doses of bOPV.”
2.Regarding the reporting of AEFI events, both vaccines were reported by same medical personnel or study areas are different place ?
Response: our study was based on the sole national AEFI surveillance system in China, which covered the whole province. All AEFI records following both IPVs were reported by the same reporters from healthcare facilities, vaccination clinics, CDC at each administrative levels, adverse drug reaction monitoring agencies and vaccine manufacturers etc.
- According to Figure-1, we can see three peaks of Salk vaccine reporting rate. Why there was the peak for reporting rate? Is there any reasons or briefly discuss about it.
Response: the supply of Salk-IPV was not always stable since it was imported. The vaccines, as well as its AEFI, would attract more attention when it was in shortage. However, we assumed it was not appropriate to state in the discussion part as it was our conjecture without any evidence from a special investigation. We will find out the real reasons for this interesting finding.
- In conclusion, the authors stated that the reporting rate and occurrence of AEFI for both vaccines were not different and the authors stated that Sabin vaccine can use safely. Why the authors describe Sabin only and not for Salk vaccine? Is there any reasons or check your conclusions and revise it.
Response: the conclusion was revised.
5.At Table-1 , the authors categorized vaccine product related reaction (minor and severe),. How did you classify and please add the operational definition as foot note in the table.
Response: we added a footnote for the description of vaccine product related reaction (minor and severe) for Table 1.
- At table -1 the authors described vaccination error. What is the meaning of vaccination error? Please add operational definition at the revised manuscript.
Response: the meaning of vaccination errors was added in the methods section.
7.At table-3, the reporting of serious AEFI event of Sabin vaccine was higher and but the consistent was not good. Is it due to over interpretation or any other reasons? Please discuss it.
Response: The possible explanation was that the reporting sensitivity of the serious AEFI following Sabin-IPV was higher due to the doubts on safety of the new vaccines from the public. However, most of the reported serious AEFI following Sabin-IPV was considered as the coincidental events after the rigorous evaluation on causal relationship.
Reviewer 3 Report
The manuscript is well written and methodology is appropriate. However, it would be of interest to know:
- Is the demographic of the families who get the Sabin government funded vaccine different to those who get the self-paid Salk strain? Could this influence reporting of AEFI?
- Why were the standard definitions of AEFI diagnosis (Brighton Collaboration) not applied in this study.
Author Response
The manuscript is well written and methodology is appropriate. However, it would be of interest to know:
1.Is the demographic of the families who get the Sabin government funded vaccine different to those who get the self-paid Salk strain? Could this influence reporting of AEFI?
Response: Thanks for you advices. All the data on AEFI and vaccination records were from the fixed or passive monitoring and reporting systems. Hence, We didn't collect the demographic information on vaccinees or their families and we could not evaluate the degree of the influence on the reporting of AEFI.
2. Why were the standard definitions of AEFI diagnosis (Brighton Collaboration) not applied in this study.
Response: Thank you for your valuable suggestions and we also were aware of the great role of Brighton Collaboration in the classification or diagnosis of the AEFI cases. However, as we stated in the limitations of this paper, the Brighton Collaboration was not widely used in China guidance on AEFI surveillance, especially at the national level.
Round 2
Reviewer 2 Report
I accept revised manuscript.